# Clinical and Genetic Findings of Autosomal Recessive Bestrophinopathy (ARB)

**DOI:** 10.3390/genes10120953

**Published:** 2019-11-21

**Authors:** Imen Habibi, Yosra Falfoul, Margarita G. Todorova, Stefan Wyrsch, Veronika Vaclavik, Maria Helfenstein, Ahmed Turki, Khaled El Matri, Leila El Matri, Daniel F. Schorderet

**Affiliations:** 1IRO-Institute for Research in Ophthalmology, 1950 Sion, Switzerland; 2Oculogenetic Laboratory LR14SP01, Hedi Rais Institute of Ophthalmology (Department B), Tunis 1007, Tunisia; 3Department of Ophthalmology, Cantonal Hospital St. Gallen, 9000 St. Gallen, Switzerland; 4Department of Ophthalmology, University of Basel, 4000 Basel, Switzerland; 5Eye Clinic, Lucerne Cantonal Hospital, 6000 Lucerne, Switzerland; 6Jules Gonin Eye Hospital, 1004 Lausanne, Switzerland; 7Department of Ophthalmology, University of Lausanne, 1004 Lausanne, Switzerland; 8Faculty of Life Sciences, Ecole polytechnique fédérale de Lausanne, 1004 Lausanne, Switzerland

**Keywords:** *BEST1*, bestrophinopathy, ARB

## Abstract

Mutations in *BEST1* cause several phenotypes including autosomal dominant (AD) Best vitelliform macular dystrophy type 2 (BVMD), AD vitreo-retino-choroidopathy (ADVIRC), and retinitis pigmentosa-50 (RP50). A rare subtype of Bestrophinopathy exists with biallelic mutations in *BEST1*. Its frequency is estimated to be 1/1,000,000 individuals. Here we report 6 families and searched for a genotype-phenotype correlation. All patients were referred due to reduced best-corrected visual acuity (BCVA), ranging from 0.1/10 to 3/10. They all showed vitelliform lesions located at the macula, sometimes extending into the midperiphery, along the vessels and the optic disc. Onset of the disease varied from the age of 3 to 25 years. Electrooculogram (EOG) revealed reduction in the EOG light rise in all patients. Molecular analysis revealed previously reported mutations p.(E35K);(E35K), p.(L31M);(L31M), p.(R141H);(A195V), p.(R202W);(R202W), and p.(Q220*);(Q220*) in five families. One family showed a novel mutation: p.(E167G);(E167G). All mutations were heterozygous in the parents. In one family, heterozygous children showed various reductions in the EOG light rise and autofluorescent deposits. Autosomal recessive Bestrophinopathy (ARB), although rare, can be recognized by its phenotype and should be validated by molecular analysis. Genotype-phenotype correlations are difficult to establish and will require the analysis of additional cases.

## 1. Introduction

Bestrophin-1 (*BEST1*) gene is associated with a wide range of ocular phenotypes, collectively termed as Bestrophinopathy. Autosomal recessive Bestrophinopathy (ARB) is part of a spectrum of retinal diseases described for the first time by Schatz et al. [1,2] as a condition caused by compound heterozygous *BEST1* mutations with a modifier effect of the first onto the second mutation. ARB is characterized by different clinical aspects including variable visual loss, hyperopia, angle-closure glaucoma, diffuse yellowish lesions corresponding to subretinal deposits merging over time, macular cysts and subretinal fluid [3]. Electrophysiological characteristics in ARB patients included reduced cones and rods responses in full-field electroretinography (ERG), with a marked decrease of light peak in EOG [2,4,5].

ARB has been hypothesized as the human “null” phenotype for Best1 [2,6]. Previous studies showed that heterozygous parents of a proband did not have any abnormal fundus findings and their EOG were normal [7,8].

Here, we report six distinct families with ARB due to mutations in *BEST1*, characterize their clinical features in great detail and show segregation as a recessive disorder.

## 2. Materials and Methods 

### 2.1. Ethics Statement

Informed consent was obtained from all family members who participated in the study. The study protocol was approved by the Hedi Rais Institute of Ophthalmology, Tunis, Tunisia; the Department of Ophthalmology, University of Basel, Basel; the Eye Clinic, Lucerne and the Jules Gonin Eye Hospital, University of Lausanne, Lausanne, Switzerland Institutional Review Board. 

### 2.2. Study Cohort 

Patients were recruited from the Hedi Rais Institute of Ophthalmology, Tunis, Tunisia; the Department of Ophthalmology, University of Basel, Basel; the Eye Clinic, Lucerne and the Jules Gonin Eye Hospital, University of Lausanne, Lausanne, Switzerland. 

A total of eleven patients with Bestrophinopathy from six unrelated families from Switzerland and Tunisia were investigated (Figure 1). Age at the time of presentation ranged between 7–44 years with a mean age of 33 years.

### 2.3. Clinical Examination of Subjects

Detailed medical history was obtained followed by clinical examination including best-corrected Snellen visual acuity (BCVA), slit-lamp examination, gonioscopy, indirect ophthalmoscopy, and fundus photography. Fundus autofluorescence (FAF) imaging with a confocal scanning laser ophthalmoscope (Heidelberg Spectralis; Heidelberg-Engineering, Heidelberg, Germany) in all eleven patients and selected family members was performed. Spectral domain optical coherence tomography (Topcon Swept source DRI OCT Triton^®^, Topcon, Tokyo, Japan) was also performed in these patients. Electrophysiologic examinations were conducted in nine patients according to the standards (both electro-oculography (EOG) and full-field electroretinography (Métrovision, France)).

## 3. Genetic analysis

### 3.1. Whole Exosome Sequencing (WES)

Peripheral blood of all subjects was collected for genomic DNA isolation from leukocytes according to standard procedures. DNA sample of the index patients was subjected to WES. Exome capture and library preparation was performed using the HiSeq Rapid PE Cluster Kit v2 with 2 μg of genomic DNA. Libraries were sequenced on a NextSeq 500 instrument (Illumina, San Diego, CA, USA). Sequence reads were aligned to the human genome reference sequence (build hg19) and variants were identified and annotated using the Nextgene software package v.2.3.5. (Softgenetics, State College, PA, USA). 

### 3.2. Variant Filtering and Analysis

To identify candidate nucleotide variants, we applied a filtering strategy: variants with an allele frequency ≤1% in either the 1000 Genome Project (1000 genomes.org) or ExAC database (exac.broadinstitute.org) were retained for further evaluation. The pathogenicity index for the identified missense variants was calculated *in silico* using PolyPhen-2 (genetics.bwh.harvard.edu/pph2) and SIFT. Given that *BEST1* is a well-established Bestrophinopathy gene, all nucleotide variants present in *BEST1* were additionally reviewed. Variants were confirmed by Sanger sequencing and segregation analysis was performed within the family.

Multiplex ligation-dependent probe amplification (MLPA) assay was performed with a SALSA MLPA probemix P367-A3 BEST1-PRPH2A (MRC-Holland, Netherlands), following the manufacturer’sinstructions; this kit contained probes for each exon of *BEST1*.

## 4. Result

### 4.1. Patients and Clinical Characteristics

Seventeen individuals, including eleven affected and six unaffected members from six families were selected for the study (Figure 2 and Figure 3; Table 1). 

#### 4.1.1. Family A

In this family, five members were affected with various severity (Figure 1; Table 1) and the pedigree indicated an autosomal recessive pattern of inheritance. The proband (II.1) was a 44-year-old female with a BCVA of 0.32 in the right eye (RE) and 0.2 in the left eye (LE) with hypermetropic refraction. Visual loss began during her third decade of life. Anterior segment examination showed bilateral shallow anterior chamber treated with YAG laser iridotomy. Fundus image revealed macular vitelliform lesions with yellow flecks and dots, extending to the mid-periphery (Figure 2A,B), fundus blue FAF showed macular hypo-FAF surrounded by markedly increased autofluorescence (Figure 2C,D). SS-OCT revealed hyperreflective accumulations on RPE, cystoid intra-retinal and serous subretinal fluid (Figure 2E,F). In full-field ERG, moderate reduced response in both scotopic and photopic conditions was observed as well as reduced light rise in EOG. 

The affected sibling (II.2) presented with a normal anterior segment with the corresponding fundus image displaying limited macular yellowish autofluorescent deposits with severe reduction in the light rise in EOG (Figure 2G–L). He was a heterozygous carrier of the *BEST 1* mutation. Three other affected brothers (patient II.3, II.4 and II.5) presented with normal fundus but with reduction in the light rise in EOG and heterozygous mutation. 

#### 4.1.2. Family B

The proband (II.3) was a 7-year-old boy (Figure 1 and Figure 2M,N) with a BCVA of 4/10 in both eyes since the age of 5 years. Proband’s fundus exhibited yellowish vitelliform deposits located in the posterior pole and ERG indicated rod-cone dysfunction. General examination revealed a pre-auricular tag at the left ear.

#### 4.1.3. Family C 

In family C, the proband (II.1) was a 17-year-old female (Figure 1) at presentation and had a history of progressive reduction of her BCVA from 0.4 in both eyes at the time of presentation to 0.16/0.25 (RE/LE) 8 years later. 

Fundus examination (Figure 2O,P) revealed multiple yellowish, vitelliform deposits in the macula and along the vessel arcades exhibiting a pseudohypopyon appearance, as well as yellowish deposits on the optic disc, showing a prominent appearance of the optic disc and better visualized on FAF (Figure 2Q,R). OCT analysis (Figure 2S,T) revealed serous subretinal fluid. ERG confirmed reduced photopic and scotopic responses (LE) and responses still in normal range (RE), whereas the mfERG was reduced centrally and paracentrally in both eyes. The corresponding EOG was reduced to 1.0/1.1 (RE/LE; normal values: 1.7–2.7). 

#### 4.1.4. Family D 

The proband (II.1) was a 28-year-old male (Figure 1) who presented with a history of progressive reduction of BCVA since childhood to 0.36 and 0.8 in both eyes (RE/LE) at the time of consultation. Fundus examination revealed multiple yellowish intraretinal deposits in the posterior pole and along the vessel arcades (Figure 3A,B) which were better visualized in the FAF image (Figure 3C,D). The OCT showed hyperreflective subretinal deposits, as well as RPE detachment affecting the macula (Figure 3E,F). The scotopic and photopic ERG responses were severely reduced as well as the EOG with a light peak: dark through of 0.88 and 1.1 (RE/LE). In addition, the patient suffered from hearing loss since the age of 2 years. Hearing loss is also mentioned in several family members. Consanguinity in the family was reported in at least 3 generations.

#### 4.1.5. Family E 

The proband (II.1) was a 13-year-old female (Figure 1) who presented with a history of reduced visual acuity and BCVA of 0.1 in RE with +3.75/−2.0/14° and 0.32 in LE with +4.75/−2.25/177°. Fundus examination revealed bilateral macular vitelliform yellow lesions and extramacular lesions along the temporal vascular arcades (Figure 3G,H), the FAF image showed autofluorescent dots along the temporal vascular arcades (Figure 3I,J). The OCT indicated subretinal deposits and subretinal fluid (Figure 3K,L). The ERG was normal while EOG was abnormal with a reduced light peak: dark through ratio.

#### 4.1.6. Family F 

The proband (II.1) from a consanguineous family of Tunisian origin currently aged 35 years (Figure 1), was 20 years old when he was first seen for reducing vision. His BCVA was 0.6 with +1.5 in RE and 0.1 with +2.5 in LE. Fundus examination revealed RPE changes at the periphery and the first diagnosis given was a RP with macular edema. Indeed (Figure 4A,B), the FAF image showed areas of hyperautofluorescence (Figure 4C,D). The OCT showed massive foveal schisis, subretinal detachment and diffuse choroidal thickening (Figure 4E,F). Full field ERG showed reduced scotopic responses and severely reduced photopic responses (30Hz Flicker) while EOG was pathological. The patient also presented with high intraocular pressure and underwent several glaucoma procedures. His visual acuity dropped to 0.1 few years later. 

His brother (II.2) first came to our hospital at the age of 22 years. He mainly complained of photophobia. BCVA at that time was 0.5 with +5,75 = −2,25/10° in RE and 0.4 with +5.75 = −1.75 in LE. Full field ERGs were mildly subnormal, while EOG were not performed. The fundus examination showed RPE alterations and the fundus autofluorescence in both eyes showed areas of hyperautofluroescence (Figure 4G,H). The OCT showed bilateral subfoveal schisis, as well as subretinal detachment and diffuse choroidal thickening (Figure 4I,J). Few years later, his BCVA dropped to 0.16 bilaterally. 

### 4.2. Exome Sequencing and Causal Variants Identification

Seven patients were found to carry homozygous or compound heterozygous variants in *BEST1* (NM_004183.4). Six previously reported mutations were identified: p.L31M in Family A, p.E35K in Family B, p.R141H and p.A195V in Family C, p.R202W in Family D, p.Q220* in Family E. One novel homozygous mutation was observed: p.E167G in Family F. 

In Family A, a homozygous substitution c.(91C > A);(91C > A) in exon 2 of *BEST1* was found in the proband resulting in a substitution of leucine at codon 31 with methionine p.(L31M);(L31M). This alteration was not reported in the 1000 Genome Project or in the ExAC database and was only recently reported in compound heterozygous state [9]. Both SIFT and PolyPhen-2 predicted this mutation to be deleterious. Three mildly affected brothers (II.2, II.3, and II.4) and the unaffected parent (I.2) were heterozygous for this mutation. The index patient underwent MLPA analysis to exclude the presence of a large genomic rearrangement and no variation was observed.

In family B, we observed a known missense mutation c.(103G>A);(103G>A), p.(E35K);(E35K) in exon 2. This variant was not reported in the 1000 Genome Project or in the ExAC database and was only reported in heterozygous state [10]. Both SIFT and PolyPhen-2 predicted this mutation to be deleterious. Consistent with the clinical findings, the p.E35K mutation was observed to be homozygous in the affected proband (II.1) and heterozygous in the unaffected parent (I.1). 

In family C, we identified p.R141H and p.A195V compound heterozygous mutations, which showed a recessive segregation pattern within the family. The p.R141H resulted from a substitution c.422G > A in exon 4 and the p.A195V mutation was a result of a substitution c.584C > T in exon 5. Both these mutations have already been described in association with ARB [2,10,11]. The unaffected mother and sister were carriers of the p.A195V mutation. 

In family D, we observed a homozygous mutation c.(604C>T);(604C>T) resulting in a substitution of arginine at codon 202 with tryptophan p.(R202W);(R202W). This variant was predicted to be deleterious by both SIFT and PolyPhen-2 and has recently been reported by Gao and al. [12] in a compound heterozygous form with p.R141H.

In family E, we detected a homozygous nonsense mutation c.(658C > T);(658C > T) in exon 5 resulting in the generation of a premature termination codon at Q220. In ExAC, the Q220* variant is present at a very low minor allele frequency (1/121400 individuals of European and African American ancestry) and was previously reported [13]. This variant was predicted to be damaging by SIFT and PolyPhen-2 and to cause loss of normal protein function either through protein truncation or nonsense-mediated mRNA decay. 

In family F, a novel homozygous substitution c.(500A>G);(500A>G) inducing the replacement of glutamic acid at codon position 167 with glycine p.(E167G);(E167G), was observed. This mutation was not present in the 1000 Genome Project or the ExAC database and was predicted to be damaging by SIFT and probably damaging by PolyPhen-2. This amino acid is very well conserved down to c. elegans and d. melanogaster. This homozygous mutation was also observed in the affected brother (II.2).

## 5. Discussion 

In this report, we analyzed the genetic and clinical characteristics of eleven ARB patients from six unrelated families. Seven variants in *BEST1* were detected, including one novel mutation. Missense mutations were the most common mutation type, which was consistent with previous studies [10,14]. Interestingly, we also observed that 4 out of 7 mutations identified in this study were located in exon 5, which is similar to other studies where 53.85% of mutations were located in exons 5 or 7 [12]. No mutations in exon 6 were identified in association with ARB [12]. 

Mutations in *BEST1* cause a wide range of ocular phenotypes which are collectively termed Bestrophinopathy. This pathology shows strong phenotypic heterogeneity with bilateral or unilateral lesions. ARB is a rare phenotype, which results from a complete lack of functional bestrophin-1 protein within the RPE. Initially, the disease may be asymptomatic or show incomplete penetrance. ARB has a wide age of onset, ranging from childhood to adulthood [3]. This trend can also be seen in our cohort where the disease onset ranged from the first to the third decade of life. 

Many studies, including this one, have attempted to determine genotype–phenotype correlations of Bestrophinopathy, but the evidence is limited [10,15]. However, we found that the homozygous *BEST1* mutation spectrum has certain clinical characteristics, the most common being extrafoveal and extramacular yellowish subretinal deposits.

The eleven patients reported in this study display key clinical features of the condition, including loss of central vision in early life, angle-closure glaucoma, subretinal and intraretinal fluid accumulation, macular and peripheral vitelliform lesions with yellow flecks but without autofluorescent yellow vitelliform lesions covering the whole macula like in vitelliform macular dystrophy, a lack of dominant mode of inheritance and abnormal electrophysiology (ERG and EOG light rise) The phenotype of our patients were compared to other publications (Table 2). Multimodal imaging can be helpful in better visualizing retinal abnormalities; OCT allows for easy identification of macular lesions, including hyperreflective accumulations on RPE, cystoid intra-retinal and serous subretinal fluid, and wide-field fundus autofluorescence helps to localize retinal abnormalities by revealing autofluorescent material [16]. 

ARB has been hypothesized as the human “null” phenotype for *BEST1* [2,6] as previous studies showed that heterozygous parents did not have any abnormal fundus findings and their EOG was normal [7,8]. Interestingly, and contrary to what is known, in family A, affected patients carrying heterozygous mutations showed reduced EOG light rise, even though normal fundus and preserved visual acuity were observed. This indicates that the mutations may induce a reduced expression of the disease. In this family, we identified a homozygous mutation p.L31M, which has recently been reported in another Tunisian family [9]. However, our patient showed a less severe phenotype than the two ARB patients presented by Chibani et al. [9]. The three affected brothers in family A were heterozygous for this *BEST1* mutation and had pathologic EOG. 

The index patient of family B carries a homozygous p.E35K mutation which has previously been reported by Tian et al. [10]. The patient in Tian et al. report presented with narrow anterior chamber and diffuse yellowish lesion with tiny yellow white spots. In our report, the phenotype of the homozygous proband showed rounded diffuse yellowish lesion with tiny yellow white spots scattered in the macula and near the inferior temporal vascular arcades. In addition, general examination revealed congenital malformation of the left ear with a pre-auricular tag which is not known to be associated with mutations in *BEST1*.

In family C we detected the presence of a p.A195V mutation which was previously reported to have a high allele frequency in Chinese and Japanese patients with ARB [12,17], indicating that it might be a hotspot for mutations in East Asian patients with ARB. However, the frequency of this mutation is not really known in Caucasian patients and its frequency should be evaluated in larger groups. This mutation is one of the most prevalent mutations among patients with compound mutations and it is associated with either a BVMD or of ARB phenotype, whereas individuals carrying the mutation at a heterozygous state alone did not show any phenotypic features of BVMD or ARB [10,11,12,18,19].

The second mutation observed in this family, p.R141H, has previously been reported as a compound heterozygous mutation in a Swedish ARB family [1,2]. Furthermore, p.R141H was also described to be the most common mutation in ARB in several unrelated families of European ethnicity [2,20,21]. Despite the fact that the two variants detected in family C were previously described, this is the first report where they are present in a compound heterozygous state. 

The mutation p.R202W observed in family D, was known to be associated with BVMD and ARB [12]. Clinically, patients carrying this mutation at a heterozygous state showed phenotypic features of BVMD [12]. The impact and importance of this variant was highlighted by *in vitro* analysis [22]. The authors showed that each of ARB-causing *BEST1* missense mutations (p.L41P, p.R141H, p.A195V, p.R202W, among others) induced reduced Cl^−^ channel activity when expressed alone but do not suppress wild-type channel activity. 

In family E, we observed a homozygous p.Q220* mutation. Q220 is a conserved residue and substitution at this position likely affects BEST1 function. The Q220* mutation has been previously reported in one case report [13] where the affected individual presented with recurrent choroidal neovascularization (CNV) exudation with progressive subretinal fibrosis event after anti-VEGF therapy [13]. This phenotype was also noticed in our index patient with bilateral macular fibrosis suggesting scarring CNV. 

In family F we observed a novel homozygous variant p.E167G in two affected brothers. This mutation was not present in ExAC and gnomAD database. It affects a highly conserved amino acid residue across species and was predicted to be deleterious and disease-causing by pathogenicity prediction tools. No genetic analysis was carried out in the parents. This mutation is in exon 5 and is most likely pathogenic. Functional analysis needs to be done to prove it and bigger cohorts need to be screened.

## 6. Conclusions

Our study provides more insight into the clinical characteristics of ARB, showing that despite different ages of onset, the patients’ phenotypes are generally similar. Furthermore, it seems that relatives of ARB patients who are carriers for the described mutations may also show signs of the wide phenotypic spectrum of ARB.

## Figures and Tables

**Figure 1 genes-10-00953-f001:**
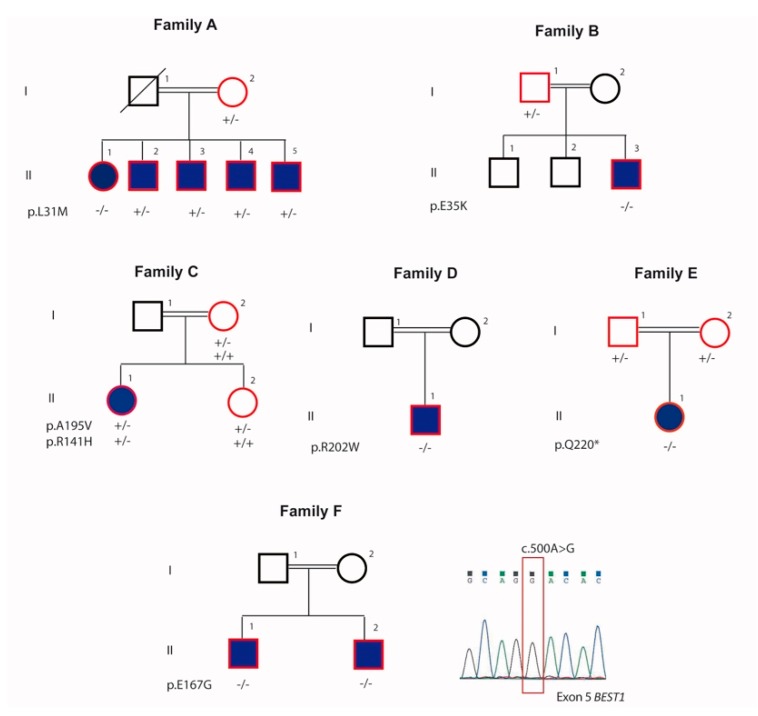
Pedigrees of the 6 families included in this study and segregation analysis of the biallelic mutations of *BEST1*. Squares represent men, and circles represent women. Solid symbols indicate patients affected with bestrophinopathy. Unfilled symbols represent unaffected family members. Symbols with red lining represent studied individuals.

**Figure 2 genes-10-00953-f002:**
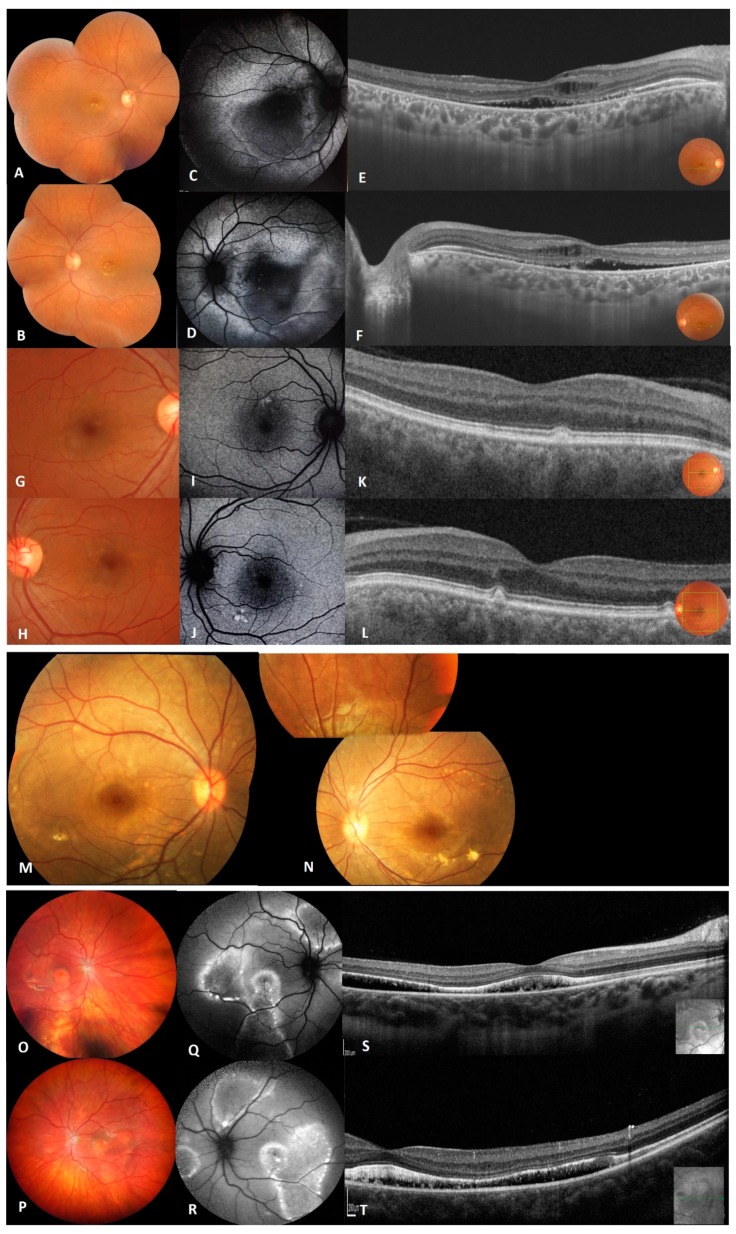
Clinical and imaging features of patients from family A (**A**–**L**), family B (**M**,**N**) and family C (**O**–**T**). (**A**,**B**) Fundus photographs of right and left eye of proband (II.1) of family A showing macular vitelliform lesions with yellow flecks and dots extending to the mid-periphery. (**C**,**D**) Fundus autofluorescence images of proband showing macular hypo-FAF surrounded by marked increased autofluorescence. (**E**,**F**) Macular OCT images of the proband showing hyperreflective accumulations on RPE, cystoid intra-retinal and serous subretinal fluid. (**G**–**J**) Fundus photographs and FAF of the sibling II.2 showing macular yellowish autofluorescent deposits. (**K**,**L**) OCT of both eyes of patient II.2 with very small focal subretinal macular deposits.(**M**,**N**) Fundus photographs of right and left eye of proband (II.3) from family B showing focal areas of sparse vitelliform deposits in the posterior pole. (**O**,**P**) Fundus photographs of right and left eye of propositus (II.1) from family C showing multiple yellowish, vitelliform deposits in the macula and along the vessel arcades with yellowish deposits on the optic disc. (**Q**,**R**) Fundus autofluorescence images of proband showing hyper autofluorescence zones along the vascular arcades and in the posterior pole. (**S**,**T**) OCT of the proband’s macula revealing bilateral diffuse flat serous subretinal fluid.

**Figure 3 genes-10-00953-f003:**
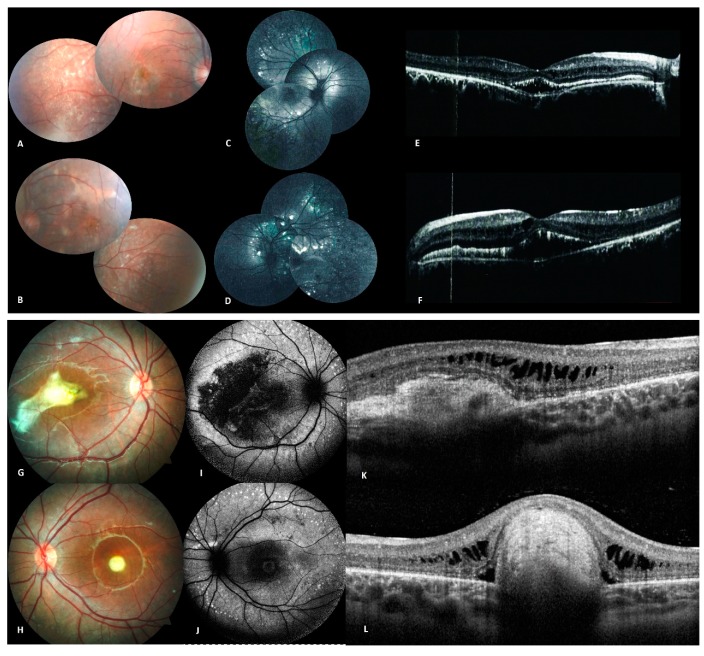
Clinical and imaging features of patients from family D (**A**–**F**), and family E (**G**–**L**). (**A**,**B**) Fundus photographs of both eyes of propositus (II.1) from family D showing multiple yellowish round vitelliform lesions, as well as subretinal fibrosis. (**C**,**D**) Autofluorescence image of both eyes showing hyper autofluorescent dots in the posterior pole and along the vascular arcades. (**E**,**F**) OCT with hyperreflective macular subretinal deposits with intra-retinal cysts. (**G**,**H**) Fundus photography of both eyes of propositus (II.1) from family E showing a macular yellow lesion with fibrosis in the right eye and a macular yellow round lesion in the left eye. (**I**,**J**) Autofluorescence image of both eyes with autofluorescent dots scattered along the vascular arcades. (**K**,**L**) OCT showing subretinal deposits with intraretinal fluid in the right eye and a subretinal mass in the left eye.

**Figure 4 genes-10-00953-f004:**
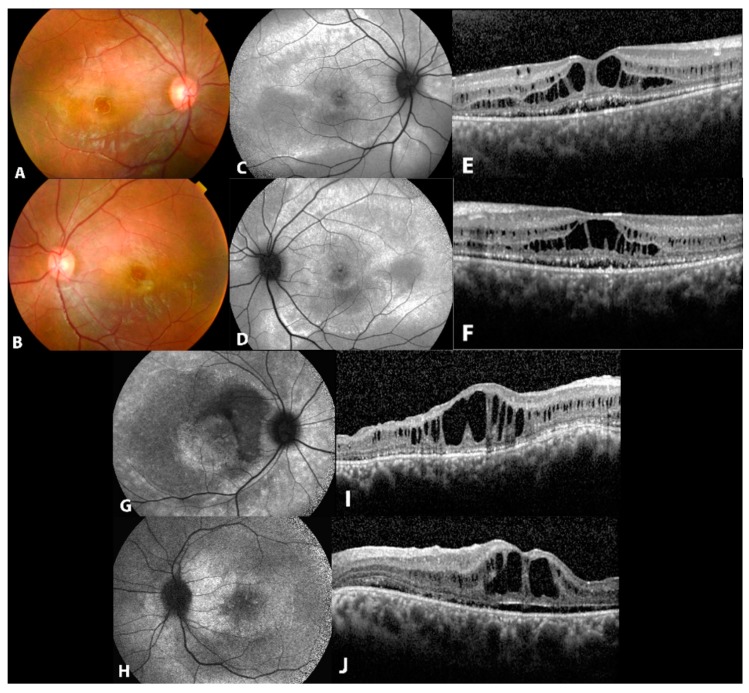
Clinical and imaging features of patients from family F. (**A**,**B**) Fundus photographs of both eyes of propositus (II.1) from family F showing macular vitelliform lesions. (**C**,**D**) Autofluorescence image of both eyes showing hyper autofluorescent areas in the posterior pole and along the vascular arcades. (**E**,**F**) OCT with subretinal fluid, macular schisis and diffuse choroidal thickening. (**G**,**H**) fundus autofluorescence in both eyes of patient (II.2) showing areas of hyperautofluroescence (**I**,**J**) OCT with subfoveal schisis, as well as subretinal detachment and diffuse choroidal thickening.

**Table 1 genes-10-00953-t001:** Summary of the clinical data.

	Age	Sex	CDVA	OCT features	ERG	EOG	Mutations	REF
**Family A**								[9]
Proband (II.1)	44	F	0.32 RE 0.2 LE	cystoid intra-retinal and serous subretinal fluid	Moderate reduced response in both scotopic and photopic conditions	Absent light peak	p.(L31M);(L31M)	
Mother (I.2)	75	F	0.4 BE	Very small sub-foveal deposits	Not done	Not done	p.(L31M);(=)	
Sibling (II.2)	50	M	0.8 BE	Very small focal subretinal macular deposits	Normal	Absent light peak	p.(L31M);(=)	
Sibling (II.3)	48	M	1.0 BE	Normal	Normal	Absent light peak	p.(L31M);(=)	
Sibling (II.4)	49	M	1.0 BE	Normal	Normal	Absent light peak	p.(L31M);(=)	
Sibling (II.5)	53	M	1.0 BE	Normal	Normal	Absent light peak	p.(L31M);(=)	
**Family B**								[10]
Proband (II.3)	7	M	0.4 BE	Not done	Not done	Not done	p.(E35K);(E35K)	
Father (I.1)	49	M	Not done	Not done	Not done	Not done	p.(E35K);(=)	
**Family C**								[11]
Proband (II.1)	17	F	RE: 0.16 LE:0.25	multiple yellowish, vitelliform deposits in the macula and along the vessel arcades exhibiting a pseudohypopyon appearance serous subretinal fluid	ERG performed on the proband confirmed reduced photopic and scotopic responses (OS) and responses still in normal range (OD), whereas the mfERG was reduced centrally and paracentrally	Reduced Arden ratio	p.(R141H);(A195V)	
Mother (I.2)	55	F	1.0, OU	Normal findings	Not done	Not done	p.(A195V);(=)	
Sibling (II.2)	18	F	1.0, OU	Normal findings	Not done	Not done	p.(A195V);(=)	
**Family D**								[12]
Proband (II.1)	28	M	RE: 0.36 LE: 0.8	hyperreflective subretinal deposits, as well as RPE detachment of retina affecting the macula	Reduced scotopic and photopic responses	Reduced Arden ratio	p.(R202W);(R202W)	
**Family E**								[13]
Proband (II.1)	13	F	RE: 0.1, LE: 0.32	Subretinal depositis and subretinal fluid	normal	Reduced Arden ratio	p.(Q220*);(Q220*)	
Father (I.1)	46	M	Not done	Not done	Not done	Not done	p.(Q220*);(=)	
Mother (I.2)	45	F	Not done	Not done	Not done	Not done	p.(Q220*);(=)	
**Family F**								Our study
Proband (II.1)	35	M	0.8 with +2 ddc	Schisis subfoveal important	Reduced scotopic and photopic responses	Pathologic	p.(E167G);(E167G)	
Sibling (II.2)	29	M	0.16 with +6 RE, 0.16 with +6 LE	Schisis, subretinal subfoveal fluid	Within normal limits	Not done	p.(E167G);(E167G)	

**Table 2 genes-10-00953-t002:** Comparing the phenotype of our families with the reported individual harboring the same mutant alleles.

	Our Study	Literature	Mutations	ACMG/AMG * Classification
**Family A**	Bilateral shallow anterior chamber macular vitelliform lesions with yellow flecks and dots, cystoid intra-retinal and serous subretinal fluid. Abnormal fundus in one heterozygous carrier. Reduction in the light rise in EOG in homozygous and all heterozygous carriers.	Normal intraocular pressure and normal anterior ocular segments in both eyes. Axial length was reduced, inflammatory vitreous cells in both eyes. Multifocal macular and extramacular involvement with yellowish deposits in the central macula for both eyes and extending to the midretinal periphery in the left eye [9].	p.(L31M);(L31M)	PM2
**Family B**	Normal interior segment, yellowish vitelliform deposits located in the posterior pole, pre-auricular tag at the left ear.	Diffuse yellowish lesion with tiny yellow white spots scattered in the macula and near the inferior temporal vascular arcade, cystoid macula edema [10].	p.(E35K);(E35K)	PM2
**Family C**	Multiple yellow vitelliform deposits in the macula and along the vessel arcades exhibiting a pseudohypopyon appearance, yellow deposits on the optic disc, serous subretinal fluid.	RPE thinning and pigment mottling in the right eye and some subretinal fibrosis in the left eye. Numerous very fine deposits anterior to the temporal vascular arcades [11].	p.(R141H);(A195V)	PS3/PS3
**Family D**	Multiple yellow intraretinal deposits in the posterior pole and along the vessel arcades, hyperreflective subretinal deposits as well as RPE detachment affecting the macula.	Multiple small, round, yellow lesions caused by vitelliform material deposits throughout the posterior pole corresponding to focal hyperreflective lesions with subretinal and intraretinal fluid [12].	p.(R202W);(R202W)	PS3
**Family E**	Macular yellow vitelliform lesions and extramacular lesions along the temporal vascular arcades.	ARB complicated by choroidal neovascularization (CNV) [13].	p.(Q220*);(Q220*)	PM2
**Family F**	Bilateral subfoveal schisis as well as subretinal detachment, RPE alterations, hyperautofluroescence delimited with hyperfluorescent ring.	-	p.(E167G);(E167G)	PM2

* ACMG/AMG classification: American College of Medical Genetics and Genomics/Association for Molecular Pathology.

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
