# Peer review of "Clinical and Genetic Findings of Autosomal Recessive Bestrophinopathy (ARB)"

_genes, 2019, doi:10.3390/genes10120953_

Round 1

Reviewer 1 Report

In the manuscript Habibi and co-authors presents their clinical and genetic findings from 6 families with autosomal recessive bestrophinopathy. The study is nicely presented. My suggestions are

Please add a table comparing the phenotype of your families with the reported individual harboring the same mutant alleles. Please add at least molecular modeling of the novel variant identified. Please add ACMG/AMG classification for all the variants identified. Figures can be combined, e.g. Make figure 1 with all the families pedigrees while the figure 2 can be clinical data in different panels. 

Author Response

Response to Reviewer 1 Comments

Point1: English language and style are fine/minor spell check required.

Response 1: we corrected typing errors

Point 2: Please add a table comparing the phenotype of your families with the reported individual harboring the same mutant alleles.

Response 2: We have added a table in the discussion (table2) comparing the phenotype of your families with the reported individual harboring the same mutant alleles.

Point 3: Please add at least molecular modeling of the novel variant identified.

Response 3: We have performed molecular modeling using chimera but didn't find any difference. However E167 is very conserved from mouse to c. elegans and d. melanogaster

Point 4: Please add ACMG/AMG classification for all the variants identified.

Response 4: These have been added to table 2.

Point 5: Figures can be combined, e.g. Make figure 1 with all the families pedigrees while the figure 2 can be clinical data in different panels. 

Response 5: The figures have been corrected.

Reviewer 2 Report

Authors made a good attempt to explain the "Clinical and genetic findings of autosomal recessive

bestrophinopathy (ARB)". Manuscript was well written with detailed explanation about bestrophenopathy. Further analysis of more cases could be helpful in establishing genotype-phenotype correlation with the  pathology.

Author Response

Response to Reviewer 2 Comments

Point1: English language and style are fine/minor spell check required.

Response1: we corrected typing errors.